# Cephalad Migration of Tunneled-Cuffed Catheter: The Importance of Post Procedure Imaging

**Asheesh Kumar \* and Ram Singh**

Department of Nephrology, Indira Gandhi Medical College, Shimla 171001, India; ramsingh304@gmail.com
\* Correspondence: asheesh03.kapil@gmail.com

**Abstract:** TCC placement is a skilled procedure; one must be aware of possible complications, particularly those related to the positioning of catheters, which are vital for the proper functioning of catheters and hemodialysis procedure. It is also equally important to be familiar with the appropriate management if such complications are encountered.

**Keywords:** dialysis; tunnel-cuff catheter; vascular access

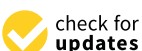



## 1. Introduction

In the developing world, despite significant improvement in awareness, counselling and networking among chronic kidney disease (CKD) patients, many patients do not have an arteriovenous fistula (AVF) for long term dialysis. Tunneled-cuffed catheters (TCC) are commonly used for CKD patients as they provide immediate access for hemodialysis and time for AVF creation. The optimal position of the catheter tip is important for adequate and uninterrupted blood flow [1,2]. As per the Kidney Disease Outcomes Quality Initiative (KDOQI) clinical practice guideline for vascular access, the proper location of the CVC tip should be at the mid-right atrium [3]. TCC insertion may be associated with complications related to its placement, which can lead to significant patient injury including central vein thrombosis, dysrhythmias, catheter malfunction and cerebral migration [3,4]. This report describes a case of malpositioned TCC superiorly into the internal jugular vein (IJV) which was diagnosed and managed successfully.

## 2. Case Presentation

A 55 year old female, with a known case of CKD-5 for 3 years, presented with uremic symptoms. The patient was obese with a BMI of 31 kg/m$^2$. The patient was started on hemodialysis through a right-sided single lumen femoral catheter. The patient was counselled for both AVF and TCC as vascular access options. In view of the intermittent hemodialysis requirement, the patient agreed for TCC insertion. A Chronic Carbothane Dual Lumen Catheter of size 14.5 F, 19 cm length, was used for this purpose with a staggered tip with side holes. The TCC was inserted in the right IJV under ultrasound guidance using the dynamic echo controlled freehand method with an out-of-plane approach for vessel puncture and long axis view of IJV for tracking the catheter under all aseptic precautions. There was no resistance for guide-wire placement during TCC insertion. A good inflow and outflow of blood through the catheter was obtained. There were no immediate procedure-related complications. As we do not have the facility of fluoroscopy or intracavitary electrocardiography (IC ECG) at our center, the patient was advised to undertake a chest X-ray after the procedure for confirmation of the catheter tip position. However, the patient had an episode of vomiting before proceeding to X-ray. Her chest X-ray film showed the TCC tip positioned superiorly, towards the head instead of downward towards the heart (Figure 1). This is a rare type of malpositioned TCC, although it is seen with central venous catheter lines [5]. The patient's catheter was taken out and the new sheath was inserted, and the catheter was reinserted into the IJV. A repeat chest X-ray showed the

catheter in the normal position. There were no other complications related to the TCC placement afterwards.

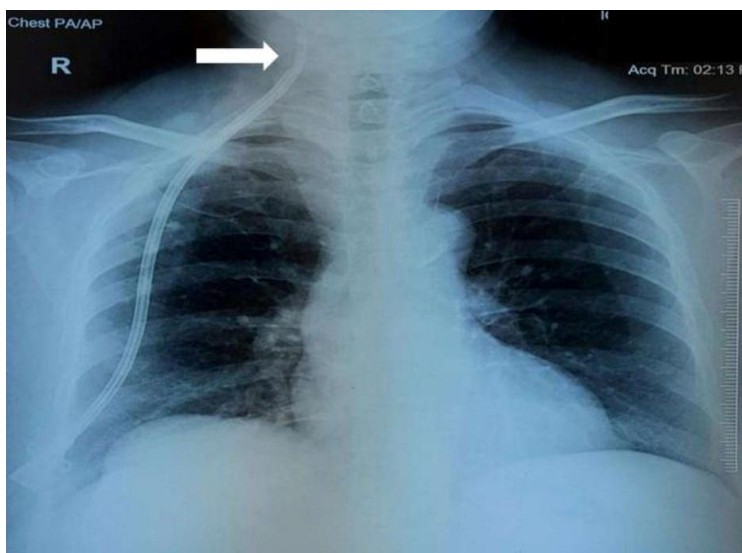

**Figure 1.** Chest X-ray film showed the TCC tip positioned superiorly, towards the head instead of downward towards the heart.

## 3. Discussion

TCC is invaluable for vascular access for hemodialysis, which is used in almost one-third of chronic hemodialysis patients [6]. Right-sided IJV is the most preferred site for TCC insertion [7]. The mechanical complications that usually arise during insertion of the catheter include hematoma, hemothorax, pneumothorax, arterial–venous fistula, venous air embolism, nerve injury, thoracic duct injury, intraluminal dissection, cardiac tamponade, and misplacement [8]. The complication rate on right-sided TCC is lower, compared with left-sided TCC [7]. Although kinking is the most common complication, malposition of the catheter is a relatively uncommon complication (incidence of 1–4% with ultrasound guidance for catheter placement), which results in catheter malfunction and requires appropriate management (re-positioning, replacement, or removal) [3,9]. As per KDOQI guidelines, TCC insertion should be image guided (ultrasound and fluoroscopic imaging) [3]. Ultrasonic guidance in combination with IC ECG monitoring may allow for a tunneled dialysis catheter to be inserted at bedside without using fluoroscopy [10]. In addition, it is sometimes possible to detect the tip of the CVC in the right atrium (RA) with the B-mode subxiphoid sonography of the heart after its insertion [11]. Post insertion imaging should be considered for detection of any malpositioning of TCC as per the Kidney Disease Outcomes Quality Initiative (KDOQI) 2019 guidelines [3].

In our patient, we used ultrasound guidance for the TCC placement, and after TCC placement there was good inflow and outflow. The sheath was directed downwards, hence, there should have been minimal chances for the catheter to migrate upwards after its placement. The patient had an episode of vomiting after the catheter placement and before the chest X-ray which might have led to an increase in intra-thoracic pressure. Additionally, there was a short segment of TCC in the internal jugular vein, as evident from the chest X-ray. Both of these contributed to catheter migration superiorly. Other risk factors for catheter migration are obesity and female gender [3,12], both of which were present in our patient. An enlarged internal jugular vein and vein valves may be another possible contributory factor, although we did not evaluate the patient for this. Catheter tip design (split tipped/stepped tipped) also influenced the outcome of the malposition.

TCC placement is a technical procedure; one should be familiar and skilled with the procedure. A central line can be left in this superiorly misplaced position [5] (depending upon its use) but TCCs should not be left in this position, as they are in place for months,

and have a high through-flow of blood in and out, which may have a detrimental impact on cerebral blood flow, although the exact estimate of this hypothesis is not known. Therefore, radiography before catheter (TCC) use is an essential requirement. It is important to be aware of the potential risk of the incorrect positioning of dialysis catheters.

## 4. Conclusions

X-ray confirmation of catheter position just after or during the catheter placement is the imperative and it should be performed in any situation.

**Author Contributions:** Conceptualization, writing—original draft preparation: A.K., writing—review and editing: R.S. All authors have read and agreed to the published version of the manuscript.

**Funding:** This research received no external funding.

**Institutional Review Board Statement:** Not applicable.

**Informed Consent Statement:** Not taken as patient identity was not disclosed.

**Data Availability Statement:** Data available on request from the authors.

**Conflicts of Interest:** The authors declare no conflict of interest.

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
