# Peer review of "Cephalad Migration of Tunneled-Cuffed Catheter: The Importance of Post Procedure Imaging"

_kidneydial, doi:10.3390/kidneydial2030039_

Round 1

Reviewer 1 Report

The Authors describe an unintentional insertion of the hemodialysis catheter cranially. This is very rare and of course, the explanation would be only hypothetical. I presume that the vein valves played a role in this complication, which should be discussed.

I do not understand the title of this report - why "not sufficient"?- if the Authors looked at the patient´s jugular vein after the procedure or more closely during the procedure, they would diagnose the malposition.

The catheter tip could be guided by X-ray or by intracardial ECG (over guidewire). This should be also mentioned.

The references are old. I suggest to the Authors to look at: PMID: 35546530, PMID: 35394395, and PMID: 34278846

Reviewer 2 Report

This case report showed the TCC migration, and suspected that the patient had an episode of vomiting after the catheter placement and before the chest x-ray which might have led to an increase in intra-thoracic pressure. I agree this speculation.

I recommend the X-R findings just after placed in the procedure for confirmation of the catheter tip position is needed to understand to this case.

Moreover, in the view of X-R, this patient suspect obesity, and related the catheter position fluctuation. If so, the body size of patient is needed. 

Round 2

Reviewer 1 Report

NA

Reviewer 2 Report

This R1 article is well modified